# Agromorphological Characterization of Quinoa (*Chenopodium quinoa* Willd.) Under Andean–Amazonian Region of Peru

**DOI:** 10.3390/plants14233689

**Published:** 2025-12-04

**Authors:** Victor-Hugo Baldera-Chapoñan, Germán De la Cruz, Segundo Oliva-Cruz, Flavio Lozano-Isla

**Affiliations:** 1Facultad de Ingeniería y Ciencias Agrarias, Universidad Nacional Toribio Rodríguez de Mendoza de Amazonas (UNTRM), Chachapoyas 01000, Peru; 2Facultad de Ciencias Agrarias, Universidad Nacional de San Cristóbal de Huamanga (UNSCH), Ayacucho 05000, Peru; 3Instituto de Investigación para el Desarrollo Sustentable de Ceja de Selva, Universidad Nacional Toribio Rodríguez de Mendoza de Amazonas (UNTRM), Chachapoyas 01000, Peru; 4Centro de Investigación e Innovación en Granos y Semillas, Universidad Nacional Toribio Rodríguez de Mendoza de Amazonas (UNTRM), Chachapoyas 01000, Peru

**Keywords:** Andean crops, *Chenopodium quinoa*, food security, genetic diversity, germplasm, plant breeding, phenotypic plasticity

## Abstract

Quinoa (*Chenopodium quinoa* Willd.) is an Andean pseudocereal of high nutritional value and remarkable phenotypic diversity, recognized as a strategic crop for food security under increasing climatic variability. In this study, the agromorphological diversity of 158 accessions cultivated in the Andean–Amazonian region of Peru was evaluated with the aim of identifying superior materials for conservation and breeding programs. The experiment was conducted using an augmented design that included three check cultivars (INIA 415 Pasankalla, INIA 420 Negra Collana, and Blanca Juli). Diversity in eleven qualitative traits was quantified using the Shannon–Weaver (H′) and Nei (He) indices, whereas twelve quantitative traits were analyzed through principal component analysis (PCA) and hierarchical clustering. The results revealed substantial intra- and inter-accession variability, with He values ranging from 0.21 to 0.76 and H′ values from 0.40 to 1.79, reflecting marked differences in growth habit, panicle morphology, stem pigmentation, and tolerance to *Peronospora variabilis* and *Epicauta* spp. Multivariate analyses identified three contrasting groups and enabled the selection of outstanding accessions, including UNTRM-367-1149, UNTRM-367-1107, UNTRM-367-1078, UNTRM-367-1079, UNTRM-367-1081, UNTRM-367-1095, and UNTRM-367-1104, characterized by high yield potential, favorable reproductive architecture, early or intermediate maturity, and low downy mildew severity. These accessions represent promising genetic resources for developing quinoa varieties adapted to transitional Andean–Amazonian environments, contributing to improved crop productivity and resilience.

## 1. Introduction

Quinoa (*Chenopodium quinoa* Willd.) is an Andean crop domesticated more than 7000 years ago in the southern Andean highlands and is now considered one of the most versatile and nutritionally valuable crops worldwide [1,2]. As a pseudocereal, it plays a fundamental role as a staple food for smallholder farmers across the Andean region and is cultivated from Colombia to Chile, with the greatest genetic diversity concentrated in Peru and Bolivia [3,4]. Its global relevance stems from its agronomic resilience and exceptional nutritional profile: quinoa grains contain high protein levels, a well-balanced composition of essential amino acids, and elevated concentrations of minerals and antioxidant compounds. These attributes position quinoa as a functional food with direct implications for food and nutrition security, particularly in vulnerable regions [5,6].

Beyond its nutritional value, quinoa exhibits remarkable phenotypic plasticity, which enables its cultivation across a wide range of agroecological conditions—from the cold, arid highlands of Peru and Bolivia to temperate and coastal environments in Asia, Europe, and North America [1,7,8]. This plasticity, supported by a broad genetic base, underlies its tolerance to salinity, low soil fertility, extreme temperatures, and limited water availability [9]. Under current climate-change scenarios, these characteristics position quinoa as a strategic crop for agricultural diversification and climate-risk reduction, thereby enhancing production stability in areas affected by environmental degradation and food insecurity [10,11].

The Andean region, considered the primary center of domestication and diversification of *Chenopodium quinoa*, preserves a remarkable agromorphological variability organized across multiple ecotypes and agroecological zones [12,13]. At the global level, there are more than 6000 quinoa accessions cultivated by farmers [14], whose genetic diversity is structured into five adaptive types: mesothermal quinoas from the inter-Andean valleys; quinoas from the northern Altiplano near Lake Titicaca, characterized by a short vegetative cycle; quinoas from the southern Altiplano salt flats, with halophytic traits and larger seeds; quinoas cultivated at sea level in central and southern Chile; and Yungas quinoas, native to subtropical areas on the eastern Andean slope [15,16,17]. However, a considerable proportion of this germplasm remains poorly evaluated under contrasting environmental conditions, which limits its use in genetic improvement programs and restricts its potential contribution to strengthening regional food security [18,19]. This situation is exacerbated by the progressive loss of traditional agricultural practices, a process that accelerates genetic erosion and reduces the species’ adaptive capacity in the face of biotic and abiotic stresses [20,21].

In this context, the northeastern Peruvian Ceja de Selva represents an emerging environment for quinoa cultivation [22]. However, there is limited systematic information on the agromorphological performance and adaptive capacity of accessions originating from high-Andean environments when grown under humid, Andean-Amazonian transitional conditions. Morphological characterization continues to be a fundamental approach for estimating genetic variability, identifying promising accessions, and selecting parental lines with desirable agronomic performance, physiological efficiency, or tolerance to biotic and abiotic stresses [4,23,24]. Such promising accessions can serve as a genetic base for the development of new varieties tailored to specific production environments, thereby contributing to resilience and long-term sustainability under changing climate scenarios [25,26].

Harnessing this diversity through the identification of outstanding accessions is therefore essential both for breeding programs aimed at improving yield and for the introduction of adapted materials into new agroecological frontiers [4,27]. In this study, a diversity panel composed of 161 quinoa accessions cultivated under Ceja de Selva conditions was evaluated with three specific objectives: (i) to characterize agromorphological diversity in both qualitative and quantitative traits, (ii) to estimate phenotypic variation and agronomic performance in the field, and (iii) to identify superior accessions with immediate potential for integration into breeding programs and for adaptive deployment in Andean–Amazonian agroecological zone.

## 2. Materials and Methods

### 2.1. Study Area

The study was conducted at the Agricultural, Livestock, and Forestry Experimental Station Lonya Chico of the National University Toribio Rodríguez de Mendoza of Amazonas (UNTRM), located in the district of Lonya Chico, province of Luya, Amazonas region, Peru (6°13′46.07″ S, 77°57′20.20″ W), at 2207 m.a.s.l., from December 2024 to July 2025. The site lies on sandy-loam soils (USDA classification) within the Peruvian Ceja de Selva, characterized by a temperate and humid climate influenced by its position on the eastern slope of the Andes (Figure 1). The mean annual temperature ranges from 17 to 20 °C, with an average annual precipitation of approximately 800 mm, concentrated mainly from November to April, followed by a dry season from May to September [22,28,29].

### 2.2. Plant Material

A total of 161 quinoa (*Chenopodium quinoa* Willd.) accessions were evaluated, comprising 158 accessions obtained from the Germplasm Bank of the National University San Cristóbal de Huamanga (UNSCH), Ayacucho, Peru (Appendix A), and three reference cultivars–INIA 415 Pasankalla (check 1), INIA 420 Negra Collana (check 2), and Blanca de Juli (check 2)–acquired from the Illpa Agricultural Experimental Station of INIA, Puno, Peru. These genetic materials, representing a broad range of Andean ecotypes, were selected to assess their agromorphological performance and adaptability under the environmental conditions of the Amazonas region.

Of the 158 accessions initially sown, 24 did not adapt to the environmental conditions of the Andean–Amazonian region and were excluded from the analyses (Appendix A).

### 2.3. Experimental Design

The experiment was established using an augmented block design, implemented through the R v4.5.1 package FieldHub v1.4.2 [30]. In the design, the 16 blocks generated in the study corresponded directly to the 16 horizontal rows of the experimental matrix composed of 16 × 13 units. Each row represented a block and included randomly one replication of the three check cultivars together with unreplicated accessions. This structure allowed the checks to function as internal controls for adjusting the environmental heterogeneity present in the field, which constituted the fundamental principle of augmented designs [31,32]. In the layout, codes “1”, “2”, and “3” correspond to the replicated check treatments used as reference standards for comparison with the non-replicated accessions. Code “1” represents INIA 415 Pasankalla, code “2” represents INIA 420 Negra Collana, and code “3” represents Blanca Juli (Appendix A). Each experimental plot measured 3 m × 5 m, with 0.10 m spacing between plants and 0.75 m between rows.

The statistical model associated with the augmented design was expressed as:
$$y_{ijk}=\mu +b_{i}+g_{j}+{\mathrm{error}}_{ijk}$$ where
$y_{ijk}$ represents the observed response for the *i*th treatment in the *j*th block and the *k*th trial within that block;
μ is the overall mean of the treatment population;
$b_{i}$ is the random effect associated with the *i*th block,
$g_{j}$ is the random effect associated with the *j*th genotype; and
${\mathrm{error}}_{ijk}$ is the random error term corresponding to the *i*th treatment, *j*th row, and *k*th trial [32].

This design was employed due to the large number of accessions to be evaluated and the limited seed availability, which required combining replicated control treatments with unreplicated lines. This structure allowed maintaining precision in comparisons among accessions and optimizing the estimation of genetic effects under field conditions [32].

### 2.4. Agronomic Management

The experimental field was prepared using conventional tillage, including plowing, marking, and furrowing to improve soil structure and facilitate uniform crop establishment. Weed control was performed manually one month after sowing, coinciding with the first fertilization. A total of 190 kg/ha of urea (46% N) and 170 kg/ha of diammonium phosphate (18% N–46% P_2_O_5_) were applied following the nutrient management protocol described by Lozano-Isla et al. [8]. Potassium was not applied due to its high availability in the soil (274 ppm), which was sufficient to meet the crop requirements. Diammonium phosphate was applied entirely during the first weeding, while urea was divided into two equal doses: the first at the initial weeding and the second during the hilling stage, which coincided with the second weeding.

Pest and disease management followed an integrated approach based on field monitoring. To control *Epicauta* sp., insecticides containing fipronil and bifenthrin were applied as foliar sprays. To manage downy mildew (*Peronospora variabilis*), fungicides containing chlorothalonil and dimethomorph were applied preventively after pest and disease assessments at 50% flowering.

### 2.5. Evaluated Variables

A total of 23 standardized descriptors for quinoa were evaluated according to the guidelines of Bioversity International et al. [33], including 12 quantitative and 11 qualitative traits. Phenological stages were recorded following the BBCH scale adapted for quinoa [34] (Appendix A). Diversity indices were estimated independently for each trait.

### 2.6. Qualitative Variables

Each qualitative variable was evaluated according to the standardized categorizations established by Bioversity International et al. [33] (Appendix A). Panicle color was determined visually at two phenological stages: 50% flowering (PCF) and 50% physiological maturity (PCM), recording the predominant color at each stage. Panicle shape (PSH) was classified according to external morphology, while panicle density (PDE) was determined by the degree of compactness. The degree of dehiscence (DD) was assessed based on natural panicle opening at maturity. Additional qualitative traits included seed coat color (SCC), growth habit (GH), main stem color (MSC), grain shape (GS), presence of *Epicauta* sp. (EPI), and lodging (LOD). All qualitative observations were made directly on the plants following uniform criteria to ensure consistency and repeatability across evaluations [35].

### 2.7. Quantitative Variables

Quantitative traits were measured directly according to Bioversity International et al. [33]. The number of days to 50% flowering (DF, days) and 50% physiological maturity (DM, days) were recorded from the sowing date. Panicle length (PL, cm) and panicle diameter (PD, cm) were measured using a measuring tape on five plants randomly selected per experimental unit to represent within-plot variability. Plant height (PH, cm) and stem diameter (SD, mm) were determined using a measuring tape and a digital caliper, respectively.

Chlorophyll content (SPAD units) was estimated at 50% flowering using a portable SPAD-502 chlorophyll meter (Konica Minolta Sensing, Inc., Osaka, Japan), taking readings from the third fully developed leaf of each plant as an indicator of photosynthetic vigor [36]. Downy mildew (*Peronospora variabilis*) severity (IPER, %) was visually assessed by estimating the percentage of infected leaf area on 10 plants per accession.

At harvest, the 1000-seed weight (TSW, g) and the seed weight from 10 plants (SW, g) were measured using a precision digital balance, while the 10 plants biomass (PB, kg) was recorded with a calibrated electronic hanging scale. The harvest index (HI) was calculated following Bioversity International et al. [33] as:
$$HI=\left(\frac{SW}{PB}+SW\right)\cdot 100$$ where SW (g) is the grain weight and PB (g) is the total above-ground biomass of the sampled plants.

### 2.8. Data Processing and Statistical Analysis

Data was recorded and managed using a combination of informatic tools for the experimental design, collect and process the data under field condition. The package Tarpuy v0.6.9 was employed for experimental planning, generation of interactive field books, and integration of activity schedules [37]. Huito v0.2.6 was used to generate reproducible QR-coded labels for efficient accession tracking and traceability [38]. The Field Book v6.2.2 application enabled systematic collection of agromorphological measurements directly in the field [39].

All statistical analyses were conducted in R software v4.5.1 [40]. For qualitative variables, genetic and phenotypic diversity were quantified using the vegan v2.7–2 package [41]. Nei’s diversity index (He) [42] was calculated to determine the number of accessions (Na), phenotypic class frequencies (fi), number of observed classes (Nc), and effective number of classes (Ne). In addition, the Shannon–Weaver diversity index (H′) [43] was estimated to assess the level of agromorphological heterogeneity among quinoa accessions.

For quantitative variables, the accessions were modeled as random effects and the check cultivars as fixed effects, in order to adjust for environmental variation and obtain unbiased genetic estimates. Variance components, broad-sense heritability (H^2^), and adjusted best linear unbiased estimates (BLUEs) were computed using the *H2cal()* function of the inti v0.6.9 package [37], which implements mixed linear models based on the lme4 v1.1-38 package [44]. The BLUEs were subjected to multivariate analyses including principal component analysis (PCA) and hierarchical clustering on principal components, using the FactoMineR v2.12 [45] and factoextra v1.0.7 [46] packages, and correlations were assessed using Spearman’s rank coefficient, implemented with the *pairs.panels()* function of the psych v2.5.6 package [47]. The analysis and the reproducible code are included in the Appendix A.

## 3. Results

### 3.1. Meteorological and Environmental Conditions for the Cultivation of Chenopodium quinoa

During the study period, daily maximum temperatures ranged from approximately 21 °C to 25 °C, whereas minimum temperatures remained mostly between 10 °C and 12 °C, with occasional drops to slightly lower values (Figure 2). The average ambient temperature fluctuated between 15 °C and 17 °C, consistently positioned between the maximum and minimum values and exhibiting limited variability. Precipitation was irregularly distributed, with numerous dry days interspersed with isolated events exceeding 5 mm and several peaks of greater intensity, some of which reached or surpassed 10–15 mm, reflecting a seasonal rainfall pattern characteristic of the climatic conditions of the area.

### 3.2. Agromorphological Diversity of Qualitative Variables

To analyze agromorphological diversity and describe phenotypic variability among quinoa accessions, eleven qualitative traits were evaluated. Phenotypic variability was quantified using Nei’s genetic diversity index (He) [42] and the Shannon–Weaver diversity index (H′) [43], which quantify heterogeneity and the range of variation among accessions. The observed values for each phenotypic class in the eleven qualitative variables evaluated are presented in Appendix A, which summarizes the complete distribution of categories recorded in the 134 accessions analyzed out of the 158 initially established.

Panicle color at 50% flowering exhibited moderate diversity (He = 0.38; H′ = 1.11), with a predominance of purple panicles (64 accessions; fi = 0.47), followed by green panicles (52 accessions; fi = 0.38), whereas red and mixed (purple–red) panicles were infrequent. Panicle shape showed higher variability (He = 0.61; H′ = 0.69); the glomerulate form was the most common (102 accessions; fi = 0.75), while the intermediate and amaranthiform types appeared in lower proportions (Figure 3a,b).

Panicle color at 50% physiological maturity was the qualitative trait with the greatest phenotypic diversity (He = 0.21; H′ = 1.79), with 10 classes identified. Yellow (48 accessions; fi = 0.35) and orange (30 accessions; fi = 0.22) were the most frequent colors, followed by pink, red, green, and various bicolored combinations in lower proportions (Table 1). Panicle density exhibited moderate variability (He = 0.70; H′ = 0.57), with the loose category predominating widely (112 accessions; fi = 0.82) over the compact and intermediate categories.

Growth habit exhibited intermediate diversity (He = 0.45; H′ = 1.01); the category “branched up to the second third” was the most common (84 accessions; fi = 0.62), followed by the simple type and the type branched up to the lower third. Main stem color showed high phenotypic richness (Nc = 9; H′ = 1.76), with green being the predominant color (57 accessions; fi = 0.42), accompanied by substantial contributions from purple and yellow tones.

The presence of *Epicauta* sp. was the most common biotic factor (87 accessions; fi = 0.64), although a considerable number of materials exhibited apparent tolerance or resistance. The degree of dehiscence showed moderate diversity (He = 0.64; H′ = 0.55), with the light category being the most frequent (103 accessions; fi = 0.76). Lodging exhibited high heterogeneity according to Nei’s index (He = 0.76), although most accessions did not present this trait (117 accessions; fi = 0.86).

The seed coat color displayed moderate diversity (He = 0.33; H′ = 1.45), with eight phenotypic categories identified. Cream-colored seeds were predominant (66 accessions, fi = 0.49), followed by white (38 accessions, fi = 0.28) (Figure 3c). The grain shape exhibited low variability (He = 0.66; H′ = 0.67), with the cylindrical type being the most frequent (107 accessions, fi = 0.79), whereas ellipsoidal and lenticular forms were less common (Table 1).

### 3.3. Agromorphological Diversity of the Quantitative Variables

The evaluation of twelve quantitative variables enabled a robust characterization of the phenotypic variation among the accessions and provided an integrated understanding of their morphological and productive performance. However, 24 accessions did not generate reliable productive information due to low germination, low initial vigor, early mortality, or an inability to complete the reproductive cycle. Only UNTRM-367-1125 exhibited a development comparable to that of the adapted accessions, although with very limited seed production and marked sensitivity to environmental conditions. Consequently, these accessions were excluded from the final analysis, including the PCA and the clustering analysis (Appendix A).

High variability was observed with the quantitative traits among the accessions. SPAD ranged from 37.5 to 82.6 SPAD units, with a mean of 58.9 units and a CV of 14.1%. IPER showed the widest interval, varying from 0 to 100.0%, with a mean of 55.9% and a CV of 57.0%, showing variable responses to downy mildew. Phenological traits showed lower relative variation: DF fluctuated between 74.0 and 94.0 days (mean = 80.0 days; CV = 6.85%), whereas DM ranged from 90.0 to 119.0 days (mean = 103.0 days; CV = 10.4%). Morphological components exhibited greater heterogeneity, with PL ranging from 9.13 to 43.5 cm (mean = 27.2 cm; CV = 22.0%) and PD from 2.74 to 15.8 cm (mean = 7.23 cm; CV = 33.3%). Plant structural traits also displayed wide phenotypic amplitudes: PH varied between 49.3 and 175.0 cm (mean = 111.7 cm; CV = 22.5%), while SD ranged from 0.10 to 1.85 cm (mean = 1.16 cm; CV = 26.1%). Yield-related variables recorded the highest levels of dispersion, with PB ranging from 0 to 3.42 kg (mean = 1.23 kg; CV = 56.0%), SW from 0 to 1091.18 g (mean = 434.3 g; CV = 55.7%), and TSW from 0.80 to 4.70 g (mean = 3.12 g; CV = 18.3%). HI ranged from 0.11 to 50.4 (mean = 27.3; CV = 33.5%) (Table 2).

The estimation of variability associated with the 16 blocks of the augmented design, together with the detailed assessment of their components, is presented in Appendix A. The results indicate that the variance attributable to block effects was low, ensuring that the BLUEs estimated for productivity-related traits remained close to the raw observed values. This reduced block influence minimized the risk of shrinkage toward the block mean and provided a more accurate representation of each accession’s performance under the experimental conditions [48].

The principal component analysis synthesized the joint variation in the correlated quantitative traits in the 134 accessions that completed their cycle under the experimental conditions. The first two components explained 51.1% of the total variability, with the first component accounting for 35.0% and the second for 16.1% (Figure 4a, Appendix A). The observed values for each quantitative trait are presented in Appendix A, providing the complete comparative basis for the characterization of the material screened.

The first component was primarily determined by PH, PB, SW, DM, DF, and SD, whose vectors were projected toward the positive end of the axis, representing a gradient of vegetative vigor and biomass accumulation. Accessions with greater height, thicker stems, heavier panicles and seeds, and later phenology were located at this end, whereas accessions with lower vigor and earlier phenology were positioned toward the negative end. The second component was mainly associated with PL, PD, and TSW, complemented by the contributions of SPAD and IPER. The positive end of the axis grouped accessions with larger panicles and greater harvest efficiency, whereas the negative end was influenced by higher severity of *Peronospora variabilis* and by lower chlorophyll content values (Figure 4a). This axis represented a productivity–health gradient.

Cluster analysis allowed the identification of three contrasting groups (Figure 4b). Group 1 included accessions with greater susceptibility to *P. variabilis*, located mainly toward the left side of the factorial plane. Group 2 comprised accessions with the highest productive potential, characterized by elevated values of PH, PL, PD, PB, and SW, and clearly concentrated in the central–upper region of the map. Group 3 consisted of accessions with low vigor, reduced biomass accumulation, and small panicles, generally associated with later phenologies and distributed toward the lower-right quadrant. These divergent patterns provide a solid basis for selecting promising accessions for breeding programs aimed at increasing yield, enhancing disease tolerance, and improving regional adaptation.

The Spearman correlation matrix (Figure 5) revealed significant associations among the evaluated variables. PB showed the strongest positive correlation with SW (r = 0.82 ***), representing the highest-magnitude association among the yield components. Likewise, PB exhibited positive correlations with PH (r = 0.63 ***), DF (r = 0.54 ***), SD (r = 0.52 ***), and DM (r = 0.56 ***). SW displayed positive correlations with PH (r = 0.52 ***), SD (r = 0.41***), DF (r = 0.30 ***), and DM (r = 0.35 ***), as well as a significant negative correlation with IPER (r = –0.43 ***). PH was positively correlated with DF (r = 0.55 ***), DM (r = 0.48 ***), SD (r = 0.58 ***), SPAD (r = 0.38 ***), and PB (r = 0.63 ***). SD showed positive associations with DF (r = 0.34 ***), DM (r = 0.31 ***), SW (r = 0.41 ***), PH (r = 0.58 ***), and PB (r = 0.52 ***).

The phenological traits DF and DM exhibited a high positive correlation with each other (r = 0.78 ***), in addition to consistent associations with PB, PH, and SD. SPAD recorded moderate correlations with DF (r = 0.32 ***), DM (r = 0.23 **), PH (r = 0.38 ***), and SD (r = 0.27 **). IPER showed negative correlations with DF (r = −0.35 ***), DM (r = −0.46 ***), PB (r = −0.44 ***), SW (r = −0.43 ***), PH (r = −0.30 ***), and SD (r = −0.31 ***), indicating that higher pathogen severity levels were associated with reductions in vegetative vigor and yield. HI displayed positive correlations with PL (r = 0.17 *), SW (r = 0.42 ***), and PB (r = 0.05), along with a negative correlation with DF (r = −0.21 *).

### 3.4. Identification of Outstanding Accessions

The identification of outstanding accessions was based on the integration of quantitative and qualitative agromorphological traits (Appendix A), complemented by the clustering structure revealed through the PCA (Figure 4b). Group 2 comprised accessions with the highest productive performance, characterized by elevated values of SW, PB, PH, PL, and PD, as well as large panicles and robust vegetative development, reflecting an architecture conducive to biomass accumulation and grain filling. Group 1 included accessions showing greater susceptibility to *Peronospora variabilis*, associated with reduced vigor and lower seed yield, whereas Group 3 included materials with reduced vegetative vigor, smaller panicles, and intermediate to low seed yield, although some individual accessions within this group stood out for either yield or tolerance.

Among the highest-yielding accessions, UNTRM-367-1149 and UNTRM-367-1107 exhibited the greatest seed yield (1091 g and 1007 g), accompanied by intermediate to tall plant heights and long panicles, positioning them in the positive region of Dimension 1, which is associated with higher productive potential (Group 2 in Figure 4b). UNTRM-367-1149 additionally displayed an amaranthiform panicle shape (PSH) and a simple growth habit (GH).

By integrating SW, HI, PH, DM, IPER, and LOD, a subset of accessions with higher agronomic potential was identified. In particular, UNTRM-367-1078, UNTRM-367-1079, UNTRM-367-1081, UNTRM-367-1095, and UNTRM-367-1104 exhibited SW values above the third quartile of the germplasm, high harvest indices, plant heights equal to or below the median, early or intermediate physiological maturity, low incidence of downy mildew (IPER ≤ 50%) (Table 2), and absence of lodging. These accessions were predominantly located within the high-performance group of the PCA (Figure 4b), which positions them as priority materials for advanced evaluation and potential incorporation into breeding programs targeting transitional Andean–Amazonian environments.

In terms of productivity gains, UNTRM-367-1079 and UNTRM-367-1081 stood out for exhibiting the highest harvest index values (HI = 50 and 40), combined with relatively short stature and short crop cycles (DM between 90 and 100 days). This combination of traits positions them as suitable accessions for the development of early-maturing cultivars that are efficient in converting biomass into yield under conditions of high climatic variability.

From a plant health perspective, several accessions exhibited high tolerance to *P. variabilis*, as evidenced by null or very low IPER values. Among them, UNTRM-367-1091 and UNTRM-367-1095 were particularly noteworthy for combining IPER = 0%, high SW values, intermediate plant stature, and absence of lodging. Complementarily, UNTRM-367-1080 and UNTRM-367-1104 exhibited good yield performance, low *Peronospora variabilis* incidence, and no damage caused by *Epicauta* sp., suggesting the presence of defense mechanisms potentially useful for selecting accessions with integrated resistance to biotic factors.

## 4. Discussion

The present study evaluated the agromorphological diversity of 158 accessions of *Chenopodium quinoa* Willd. originating from the Ayacucho region, together with three control cultivars from the Altiplano of Puno. This research represents the first introduction of quinoa into the Andean–Amazonian agroecological zone, a context characterized by climatic variability, irregular rainfall patterns, and progressive soil degradation, all of which compromise the stability of traditional crops [29]. The results demonstrated a marked phenotypic divergence among the accessions, reflected in the broad variation in panicle color and shape, inflorescence density, and growth habit, which exhibited high category richness under the agroecological conditions of Lonya Chico. The analysis of quantitative traits revealed consistent associations between growth and productivity, highlighting the relationship between plant biomass and yield, as well as the positive correlations with plant height, stem diameter, and panicle dimensions. These patterns enabled the identification of a group of accessions with greater vigor, favorable architecture, and high productive potential, including materials with voluminous panicles, moderate stature, high harvest indices, and low incidence of *Peronospora variabilis* and *Epicauta* sp. Such accessions constitute valuable resources for conservation and breeding programs aimed at strengthening the adaptation and productivity of quinoa in specific regions of Peru.

### 4.1. Diversity of Qualitative Variables

The analysis of the agromorphological diversity of the *Chenopodium quinoa* Willd. accessions revealed a high level of phenotypic variation among the 12 qualitative characters evaluated, with mean genetic diversity values (He = 0.50) and a Shannon and Weaver index (H′ = 0.97) (Table 1) that indicate substantial heterogeneity within the collection. This variation is consistent with studies demonstrating the phenotypic plasticity and diversity of quinoa, reflecting the interaction between its genetic basis and the local edaphoclimatic conditions [4,14,49,50].

Among the traits evaluated, panicle color at 50% maturity exhibited the greatest diversity, displaying tonalities ranging from yellow and orange to pink and green. This pattern is consistent with morphological assessments conducted in tropical and temperate environments [4,51] and reflects the significant influence of both genetic background and environmental conditions on the synthesis of pigments such as betalains and flavonoids, whose functional variation has been demonstrated through metabolomic studies conducted on large germplasm collections [50].

The shape and density of the panicle, dominated by glomerulate and lax types, are also consistent with reports for accessions exhibiting broad geographic adaptation [4,14,52]. The variation in this trait reflects the marked morphological divergence among the accessions, as panicle density is described as a highly variable descriptor across different genetic backgrounds and one that is sensitive to local environmental conditions [4,49].

The variation in stem coloration observed among the evaluated accessions is consistent with the patterns described in studies on morphological diversity in quinoa. Previous research has documented that stem pigmentation exhibits considerable diversity among accessions and represents a useful trait for their differentiation, both in local collections and in broader diversity panels [4,23,53]. More recently, large-scale genomic and phenotypic analyses have confirmed that this chromatic heterogeneity is a stable component of the characteristic phenotypic variability of the species and reflects the interaction between the genetic background and environmental plasticity [54]. Likewise, the variation observed in grain color and shape is consistent with the metabolic diversity reported in broad germplasm collections [50], where differences in betalains, phenolics, lipids, and specialized metabolites contribute to defining differentiated profiles of nutritional value and adaptive potential.

The frequent presence of *Epicauta* sp. enabled the identification of accessions with apparent tolerance, suggesting defensive mechanisms possibly associated with saponin production. This is consistent with studies indicating that genetic variability promotes the accumulation of secondary metabolites with protective functions against both biotic and abiotic stresses [55,56].

The phenotypic diversity recorded in this study is consistent with evaluations conducted in other regions, where quinoa exhibits marked variation in coloration, architecture, and reproductive morphology, as documented in northwestern Europe [14] and in South American collections characterized by broad morphological heterogeneity [4]. This concordance supports the notion that the qualitative differentiation observed reflects a diverse genetic basis, consistent with studies that relate phenotypic variation to germplasm origin and adaptive processes [57], as well as with genomic evidence documenting pronounced genetic structuring in the species [54].

### 4.2. Diversity of Quantitative Variables

The evaluation of quantitative variables revealed clear patterns of divergence among accessions, reflected correlations among vigor, phenology, health status, and productivity, as well as in the ordered structure identified through PCA and cluster analysis (Figure 4, Appendix A). The vigor–biomass gradient and the productivity–health axis synthesized most of the observed variation, allowing the distinction of high-yielding accessions, low-vigor materials, and accessions with greater susceptibility to *P. variabilis*. This breadth of quantitative responses is explained by the joint influence of the germplasm’s genetic structure and the environmental conditions that modulate vigor, phenology, and yield across accessions.

PB showed the strongest positive correlation with SW (r = 0.82 ***), confirming that the accumulation of reproductive biomass constitutes a central axis in determining yield. This pattern is consistent with studies demonstrating that panicle biomass and numerical yield components account for a substantial proportion of the productive variability in quinoa [51,58], as well as with comparative analyses in which variation in panicle weight and seed number has been key to differentiating accessions across multiple environments [4,57].

Structural vigor exerted a decisive influence on productivity. PB showed positive associations with PH (r = 0.63 ***), SD (r = 0.52 ***), DF (r = 0.54 ***), and DM (r = 0.56 ***), while SW correlated with PH (r = 0.52 ***), SD (r = 0.41 ***), DF (r = 0.30 ***), and DM (r = 0.35 ***). These patterns indicate that greater vegetative stature contributed to enhanced light interception and to the development of reproductive structures of larger volume, consistent with studies reporting that accessions with more vigorous growth accumulate higher biomass and achieve greater yields [51,57,58]. However, the magnitude of these effects can be modulated by environmental conditions. It has been documented that nocturnal heat stress reduces total biomass and alters the partitioning of assimilates toward reproductive organs, thereby decreasing grain formation and filling, which confirms the sensitivity of yield to constraints on dry matter accumulation under adverse conditions [59,60].

Phenological traits were consistently integrated into the observed patterns. DF and DM showed a strong positive correlation with each other (r = 0.78 ***) and were closely associated with PB, PH, and SD, suggesting that a slightly extended growth cycle favored biomass accumulation prior to panicle differentiation. This behavior aligns with observations in low-altitude environments, where longer growth periods enhance grain filling and increase productivity [59]. However, this trend may be reversed in cold or humid regions, where extended cycles expose the crop to climatic risks during grain filling, thereby reducing yield, as reported in western Europe [14].

SPAD showed moderate associations with PH (r = 0.38 ***), SD (r = 0.27 **), DF (r = 0.32 ***), and DM (r = 0.23 **), indicating that the preservation of foliar functionality contributed to structural vigor. This pattern is consistent with studies in which the maintenance of photosynthetic activity during grain filling has been fundamental for sustaining productivity [58].

Downy mildew severity (IPER) showed consistent negative correlations with DF (r = −0.35 ***), DM (r = –0.46 ***), PB (r = –0.44 ***), SW (r = −0.43 ***), PH (r = –0.30 ***), and SD (r = −0.31 ***). These results indicate that high levels of infection simultaneously affected phenology, vegetative architecture, and grain production. The literature confirms that *Peronospora variabilis* reduces growth, reduces vigor, and limits reproductive development in susceptible cultivars, particularly under conditions of high humidity [16,56]. Finally, HI showed positive correlations with PL (r = 0.17 *), SW (r = 0.42 ***), and PB (r = 0.05), as well as a negative correlation with DF (r = –0.21 *). This variability reflects the physiological complexity of assimilate allocation, a pattern that has been widely reported in studies on genetic diversity and multilocational stability [4,49].

### 4.3. Selection of Promising Accessions

Motivated by the need to diversify agricultural production and strengthen food security in Andean–Amazonian ecoregion—where traditional crops are affected by climatic variability and environmental degradation—an appropriate quinoa ideotype for this region should combine high yield, low to intermediate plant stature to reduce lodging, and earliness. Additionally, tolerance to pests and diseases typical of humid environments is required, together with a simple stem and amaranthiform panicles, whose open structure enhances inflorescence ventilation and reduces insect incidence [8]. This set of characteristics defines the desirable profile for identifying promising accessions with potential for adaptation and use in breeding programs targeting Andean–Amazonian ecoregion.

In the present study, the comparisons made among accessions demonstrated the relevance of these selection criteria. When contrasting the yields obtained here with those reported in other environments, it was observed that while Lozano-Isla et al. [8] documented a maximum of 533 g per ten plants under inter-Andean conditions, and Vleugels et al. [14] recorded up to 1370 g per ten plants in the temperate and humid environments of northwestern Europe, several of the accessions evaluated in this study exceeded 1000 g per ten plants. This differential performance confirmed that the traits used as selection criteria—plant biomass, inflorescence dimensions, structural robustness, phenological adjustment, and lower incidence of *Peronospora variabilis*—operate as key determinants for the expression of productive potential under warm and humid conditions.

Multivariate analysis allowed the grouping of the most productive accessions within a set defined by greater reproductive biomass, voluminous panicles, and robust stems—attributes that reduce the risk of lodging and enhance yield stability [4,58,61,62]. In addition, amaranthiform panicles stand out because their open structure promotes inflorescence ventilation and reduces pest incidence [8].

The combination of yield, harvest index, plant height, physiological maturity, low incidence of *P. variabilis*, and absence of lodging allowed the identification of a subset of materials with greater agronomic potential. These accessions integrate low or intermediate stature, early or intermediate cycles, and efficient biomass accumulation in the plant—traits consistent with studies conducted at low and mid altitudes in Asia [3] and relevant for short-season environments [63,64]. In contrast, accessions with very late cycles tend to show reduced yield in temperate and humid environments [14], underscoring the importance of phenological adjustment in warm regions.

From a plant health perspective, several accessions exhibited high tolerance to *P. variabilis*, expressed as null or low incidence values—an important finding given the impact of downy mildew on vigor and panicle size [16]. Some accessions also showed no damage caused by *Epicauta* sp., which is consistent with transcriptomic and metabolic mechanisms such as saponin-mediated defense [55,65].

From 158 accessions used in this study, 24 were not able to adapt to these new environmental conditions—a pattern consistent with observations in tropical and subtropical environments, where early establishment strongly depends on the interaction between ecotype and micro-environmental conditions [35,49]. Within this group, the accession UNTRM-367-1125 showed an atypical behavior: it developed normal vegetative growth but did not produce grain. Such reproductive failures have been reported in Andean accessions evaluated outside their ecological range, where photoperiod sensitivity and the mismatch between flowering and environmental conditions limit fruit set even in the absence of extreme temperatures [58]. Studies conducted in warm and semi-arid regions further confirm that flowering is a particularly vulnerable phase to moderate thermal increases [16], supporting the interpretation that the absence of grain in UNTRM-367-1125 reflects high reproductive sensitivity under the warm and humid conditions of the experiment.

### 4.4. Limitations and Future Perspectives of the Study

This study constitutes the first analysis of a quinoa diversity panel in the Amazonas region and provides an initial overview of the agromorphological variation and adaptive behavior of *C. quinoa* under the agroecological conditions of Lonya Chico, Peru. The evaluation was carried out in a single environment and growing season because the material corresponded to an initial introduction and seed availability was limited, which limited the use of fully replicated designs. Additionally, the hilling practice applied reduced lodging incidence, partially attenuating differences among less stable accessions; nevertheless, identifying lines with low lodging susceptibility remains strategically important for reducing management costs for farmers. Furthermore, the high-Andean cultivars used as checks did not express their productive potential, confirming marked ecotypic differences between materials from Puno and Andean–Amazonian environments.

Despite these limitations, the results lay the foundation for continued germplasm characterization, the integration of molecular analyses, and the evaluation of nutritional attributes aimed at breeding. Subsequent stages will involve multi-site or multi-year trials to deepen the understanding of phenotypic diversity, quantify genotype × environment interactions, and validate the stability of the most promising accessions. The inclusion of native accessions from the inter-Andean valleys of Cusco will broaden the genetic base and facilitate the identification of populations better adapted to the Andean–Amazonian region. Complementarily, the integration of grain-quality and nutritional assessments will be essential to guide breeding programs focused on sustainability, resilience, and the comprehensive utilization of quinoa in this region.

## 5. Conclusions

The agromorphological characterization of *Chenopodium quinoa* accessions in the Amazonas region revealed substantial intra- and inter-accession variability, reflected in the phenotypic diversity of both qualitative and quantitative traits. Positive correlations among plant height, panicle dimensions, biomass accumulation, and grain yield highlighted the central role of vegetative architecture in determining productive potential. Multivariate analyses also identified accessions with superior agronomic performance, enhanced tolerance to *Peronospora variabilis* and *Epicauta* sp., and high production capacity. These materials represent strategic resources for breeding programs aimed at improving the crop’s adaptation and resilience in the warm and humid environments of the Andean–Amazonian region. The results provide a foundation for establishing regional germplasm banks and conserving quinoa’s genetic diversity, reinforcing its importance as a nutritious and adaptable crop for sustainable agricultural systems.

## Figures and Tables

**Figure 1 plants-14-03689-f001:**
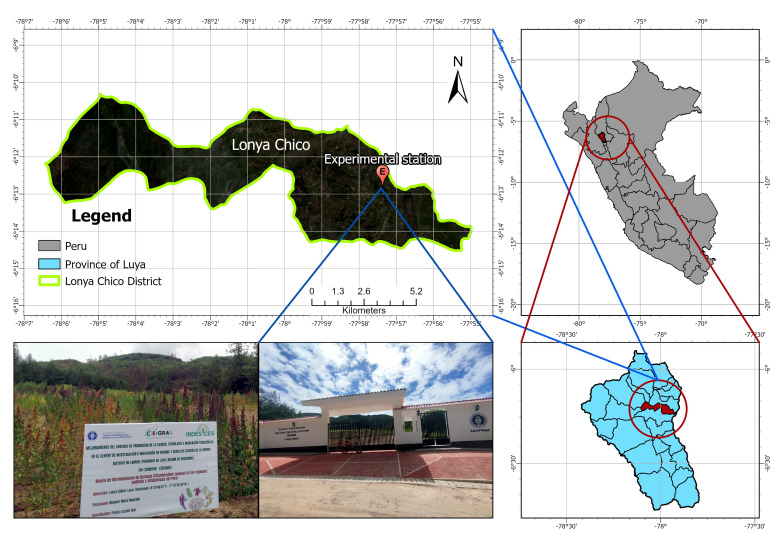
Location map of the Agricultural, Livestock, and Forestry Experimental Station Lonya Chico. The site belongs to the Andean–Amazonian agroecological zone and served for the characterization of 161 quinoa (*Chenopodium quinoa* Willd.) accessions under field conditions.

**Figure 2 plants-14-03689-f002:**
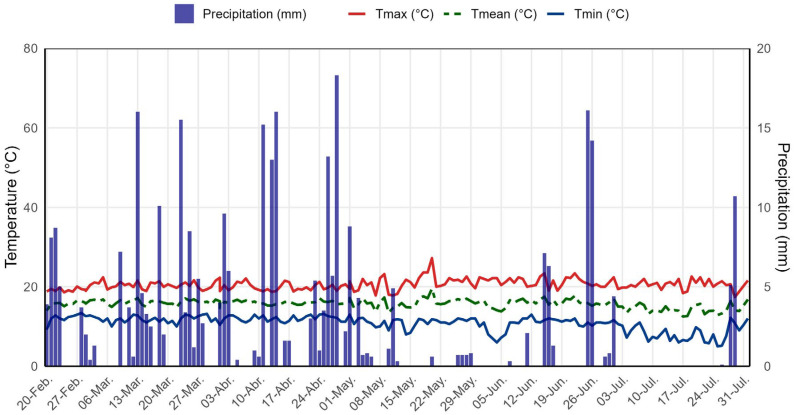
Daily variation in temperature and precipitation recorded at the experimental site in the Andean–Amazonian zone of the Amazonas region, Peru, during the quinoa field trial conducted from February to July 2025.

**Figure 3 plants-14-03689-f003:**
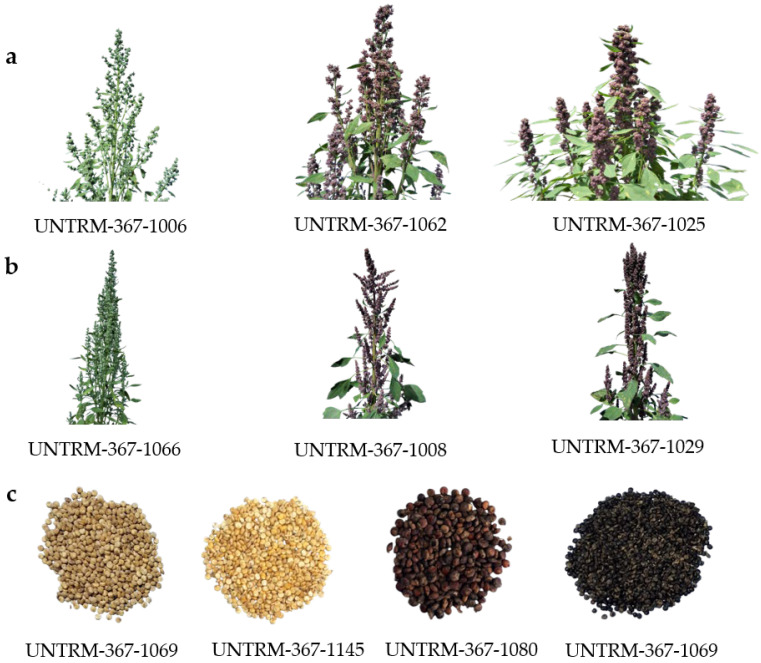
Phenotypic diversity observed in panicle traits and seed episperm color within our panel of quinoa accessions: (**a**) Glomerulated panicles; (**b**) Amaranthiform panicles; (**c**) Seeds episperm color variation.

**Figure 4 plants-14-03689-f004:**
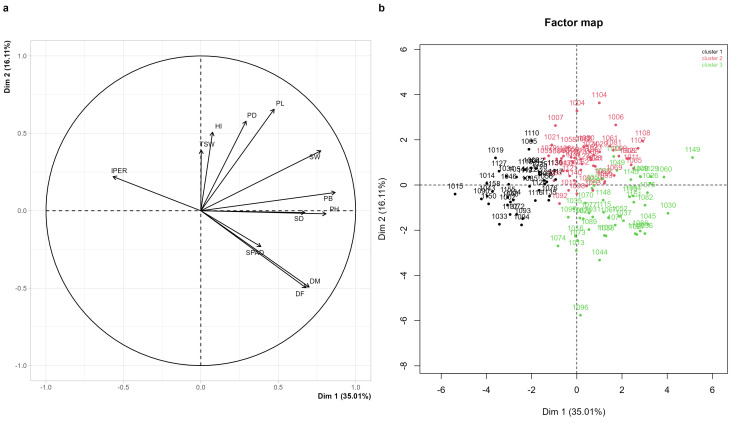
Principal component analysis (PCA) and hierarchical clustering of 158 quinoa accessions (*Chenopodium quinoa* Willd.) evaluated at the Agricultural, Livestock, and Forestry Experimental Station of Lonya Chico, Amazonas, Peru. (**a**) Correlation and relationships among quantitative variables. (**b**) Factor map for individual for hierarchical clustering. Where panicle length (PL, cm), panicle diameter (PD, cm), number of days to 50% flowering (DF, days), number of days to 50% physiological maturity (DM, days), 1000-grain weight (TSW, g), biomass of 10 plants (PB, kg), seed weight from 10 plants (SW, g), chlorophyll content at 50% flowering (SPAD units), plant height (PH, cm), stem diameter (SD, mm), downy mildew severity caused by *Peronospora variabilis* (IPER, %), and harvest index (HI).

**Figure 5 plants-14-03689-f005:**
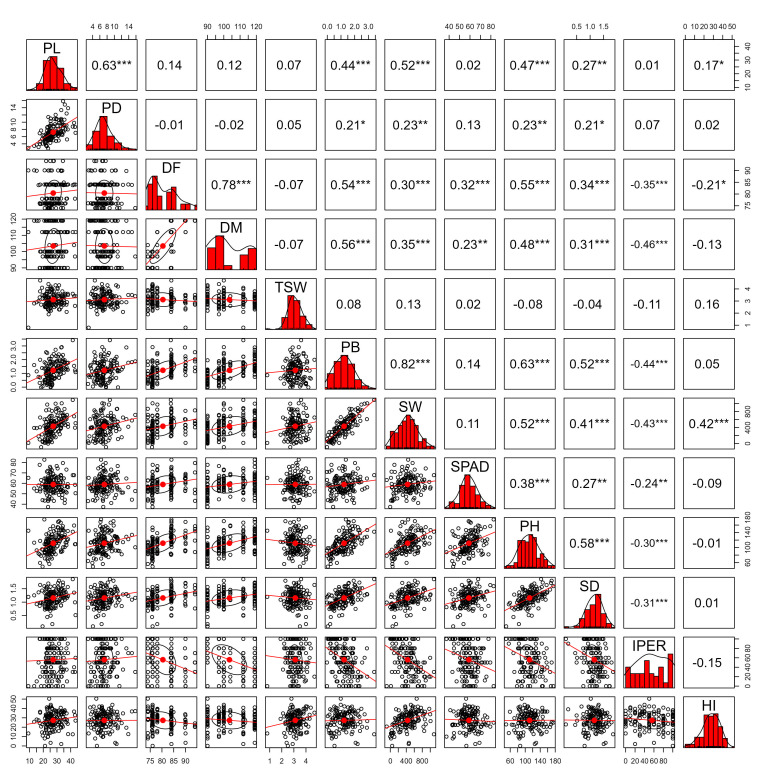
Spearman correlation matrix among morphological and productive variables evaluated in 158 quinoa (*Chenopodium quinoa* Willd.) accessions. The analyzed quantitative traits included panicle length (PL, cm) and panicle diameter (PD, cm), number of days to 50% flowering (DF, days) and to 50% physiological maturity (DM, days), 1000-grain weight (TSW, g), biomass of 10 plants (PB, kg), seed weight from 10 plants (SW, g), chlorophyll content at 50% flowering (SPAD units), plant height (PH, cm), stem diameter (SD, mm), downy mildew severity caused by *Peronospora variabilis* (IPER, %), and harvest index (HI). Significance levels: *p* < 0.05 (*), *p* < 0.01 (**), *p* < 0.001 (***).

**Table 1 plants-14-03689-t001:** Agromorphological diversity and qualitative characterization of 158 quinoa (*Chenopodium quinoa* Willd.) accessions evaluated at the Agricultural, Livestock, and Forestry Experimental Station of Lonya Chico, Amazonas, Peru (2025).

Qualitative Trait	Phenotypic Classes Observed	Na	fi	Nc	Ne	He	H′
Panicle color at 50% flowering	Mixture (purple and red)	9	0.07	4	2.64	0.38	1.11
	Purple	64	0.47				
	Red	11	0.08				
	Green	52	0.38				
Panicle shape	Amaranthiform	7	0.05	3	1.65	0.60	0.69
	Glomerulate	102	0.75				
	Intermediate	27	0.20				
Panicle color at 50% physiological maturity	Yellow	48	0.35	10	4.71	0.21	1.79
	Orange	30	0.22				
	White	3	0.02				
	Gray	1	0.01				
	Brown	2	0.01				
	Purple	10	0.07				
	Red and yellow	1	0.01				
	Red and pink	12	0.09				
	Pink	19	0.14				
	Green	10	0.07				
Panicle density	Compact	6	0.04	3	1.43	0.70	0.57
	Intermediate	18	0.13				
	Loose	112	0.82				
Growth habit	Branched with undefined main panicle	4	0.03	4	2.25	0.44	1.01
	Branched up to the second third	84	0.62				
	Branched up to the lower third	24	0.18				
	Simple	24	0.18				
Main stem color	Yellow	22	0.16	9	4.2	0.24	1.76
	Orange	7	0.05				
	White	7	0.05				
	Gray	3	0.02				
	Brown	5	0.04				
	Purple	21	0.15				
	Red	7	0.05				
	Pink	7	0.05				
	Green	57	0.42				
Presence of *Epicauta* sp.	No	49	0.36	2	1.86	0.54	0.65
	Yes	87	0.64				
Degree of dehiscence	Light	103	0.76	3	1.59	0.63	0.59
	Regular	32	0.24				
	Strong	1	0.01				
Lodging of the plant	No	117	0.86	2	1.32	0.76	0.40
	Yes	19	0.14				
Seed coat (episperm) color	White	38	0.28	9	3.07	0.33	1.45
	Brown	4	0.03				
	Light brown	3	0.02				
	Dark brown	3	0.02				
	Reddish brown	12	0.09				
	Cream	66	0.49				
	Black	4	0.03				
	Transparent	5	0.04				
		1	0.01				
Grain shape	Cylindrical	107	0.79	4	1.55	0.65	0.67
	Ellipsoidal	7	0.05				
	Lenticular	21	0.15				
		1	0.01				
**Mean ± SE**				**4.82 ± 0.90**	**2.39 ± 0.35**	**0.50 ± 0.06**	**0.97 ± 0.15**

**Note.** Total number of accessions evaluated (Na); frequency of phenotypic classes (fi); number of observed classes (Nc); effective number of classes (Ne); genetic diversity index (He); Shannon–Weaver diversity index (H′); standard deviation (SE).

**Table 2 plants-14-03689-t002:** Descriptive statistics of the quantitative variables evaluated in the 158 quinoa (*Chenopodium quinoa* Willd.) accessions.

Quantitative Trait	Unit	Mean	Min	Max	CV(%)
Chlorophyll content at 50% flowering	SPAD	58.9	37.5	82.6	14.1
Incidence of *Peronospora variabilis*	%	55.9	0.0	100.0	57.0
Days to 50% flowering	days	80.0	74.0	94.0	6.9
Days to 50% physiological maturity	days	103.0	90.0	119.0	10.4
Panicle length	cm	27.2	9.13	43.5	22.0
Panicle diameter	cm	7.23	2.74	15.8	33.3
Plant height	cm	111.7	49.3	175.0	22.5
Stem diameter	cm	1.16	0.10	1.85	26.1
Biomass of 10 plants	kg	1.23	0.00	3.42	56.0
Seed weight from10 plants	g	434.3	0.00	1091.2	55.7
1000-grain weight	g	3.12	0.80	4.70	18.3
Harvest index	ratio (%)	27.3	0.11	50.4	33.5

Where mean (Mean), minimum (Min), maximum (Max), and coefficient of variation (CV%) correspond to the quantitative variables evaluated in the 158 accessions that completed their phenological cycle in the Andean–Amazonian region.

## Data Availability

The dataset, code, and reproducible analyses are included in the Appendix A. The full code and analytical workflow are available in the public GitHub repository: https://github.com/Victor-Hugo-Baldera/Quinoa_diversity_vh.git (accessed on 26 November 2025).

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
