# Peer review of "Agromorphological Characterization of Quinoa (Chenopodium quinoa Willd.) Under Andean–Amazonian Region of Peru"

_plants, 2025, doi:10.3390/plants14233689_

Round 1

Reviewer 1 Report

Comments and Suggestions for Authors

Dear editor, 
Dear authors,

I believe this manuscript is generally well-written and contains interesting information. English language is very good and the paper is written clearly and understandable. 
However, the manuscript must be improved before it can be accepted for publication. 

This was a trial with augmented design in only 1 trial season; fully replicated design would have been better. Variation between blocks should be assessed and discussed. Another major remark is that the actual results obtained in this study for each accession are not presented. Without presenting this information, the significance of the manuscript is low, and the manuscript should in my opinion be rejected. Promising accessions should be mentioned. 

Below you can find my detailed comments. Please address each comment. Additional small textual edits can be found in the attached document. 

General
*********

I have a problem with the numerous abbreviations that are used in this study. A lot of abbreviations are defined that are not always so straithforward. For the reader it is impossible to remember all abbreviations. However, abbreviations are often written in full again at multiple places in the text. E.g. productivity traits are repeated have been introduced in the M&M and are repeated in line 234-244, and again in line 265-275. Also they are repeated in figures and Tables. Do we really need all these abbreviations then? I suggest to limit the abbreviations and write them in full as much as you can. 

Introduction
*************

 - There are 5 types of quinoa, including the Altiplano type, valley type and the Yungas type. Given the environment if the study site, I would expect that yungas type or valley type performs best in this region. It would be good to add this information in the introduction. You can then use it again in the discussion to explain why some of the check cultivars (Altiplano types) did not perform well in this environment (line 428 - 430). 

 - line 58-59: 2x erosion in the same sentence.  Try to find a different phrasing, e.g. loss of traditional cultivation practices

 - line 89-91: please provide total annual precipitation (mm) for the trial region. 

 - One of the objectives of this study is to identify accessions with potential for breeding in de Andean-Amazonian region (line 78-80). This implies that you need to describe what traits are important for that region. What kind of varieties are farmers growing in this region now, and what would we need for this region? Do the farmers in that region need early or late maturing varieties? tall plants or short? big os small seeds? Is susceptibility to downy mildew an issue? … Please describe which traits a suitable variety for that region would have. Then later in the results section you can indicate accessions that meet those criteria. 

M&M
****
 - line 93-96 : Do you include in your manuscript a (Supplementary) Table with identification of the plant material (accession name, UNSCH reference number, accession properties) ? That would be an added value for the paper. 

 - Line 100: a trial plan figure with organisation of the trial would be nice. Where were the control accessions placed in every block? Name your blocks from 1 to 16.

 - line 120-122: please provide total units of NPK fertilization/ If I calculate well, the trial received 118 units N, 78 units P and 0 units K units. N and P fertilization seems very adequate to me. Why no potassium fertilization? Does quinoa not use potassium?! 

 - line 161: For small-seed crops like quinoa, 1000-seed weight (TSW) is more commonly used than 100-seed weight. The used method to weight 1x 100 seeds is OK, but I strongly recommend to present the data as 1000-seed weights (just multiply by 10). The numbers will then also be larger than 1, making them easier to present. Use the abbreviation TSW instead of W100: TSW is commonly used. 

 - line 162: I suggest to define easier abbreviations for SW10P and W10P. it is well described in the M&M that these traits were obtained using seeds from 10 plants, it is not necessary to include this information in the abbreviation. You may as well use 'SW' for seed weight and 'PAN' for panicle biomass. It will be easier to understand for the reader. 

 - line 189: this manuscript contains a lot of abbreviations. Abbreviations are useful only when they are used many times as they shorten the text. This abbreviation HCPC is only used once: just write it in full then please. Please also check for all other abbreviations in the text: if they are used only 3 times or less, just write them in full. 

Results
*********
 - The trial was organized in different blocks. The variability between the 16 blocks must be presented. You can again refer to the figure in the M&M.  In the discussion, you should then discuss the variability between blocks. Ideally, variation is low. In that case, BLUEs for the productivity traits for all accessions are close to the actually observed value. In case of high variability between the blocks, we can expect large adjustments to the BLUEs, which often causes distortion (shrinkage to the block mean) in productivity traits. In that case, the productivity traits obtained in your study for the accessions should be interpreted more carefully. This must be assessed and discussed. 

 - It would be an added value if you could present the weather conditions (daily average temperature and daily precipitation for the growing seasons during the trial, and compare this to the long term average. This would allow you to discuss if the trial year was a dry, normal or wet year, and if it was warm, normal or cold. This is important information to explain why some accessions performed well or less well than expected. 

 - A major remark is that the actual results obtained in this study for each accession are not presented. I expect to see 1 or 2 (Supplementary) Table(s) with the results for all the 12 quantitative and 11 qualitative traits for each accession. This is important information for readers that are searching for an accession with a specific trait to use in their own study. Without presenting this information, the manuscript has only little value to the broader audience and should in my opinion be rejected.  

 - A table is presented with the general findings for the 11 qualitative traits observed in this study. You also need to make a similar Table in the main document with the results for the quantitative traits: min and max for each trait, average, coefficient of variation (CV%), … This will help the reader to grasp the diversity present in the screened set, and compare the values obtained in your study to other quinoa screening studies. E.g. was the seed yield, plant height, ... obtained in your study higher or lower than in other studies? Now there is no way for the reader to assess this. 

 - Line 234 - 244: abbreviations only need to be defined the first time they are introduced in the text. All abbreviations here already have been introduced in the M&M. No need to do it again here. Idem line 265-275. For figures and Tables (eg Figure 3) it is better to define the abbreviations again in the legend, as you do. 

 - line 246-248: 1 digit after decimal point is enough for the % of variance explained: 35.1% and 16.0%. 

 - line 291: (1,103.35 g and 1,010.82 g). Do we need the digits after the decimal point?? --> I suggest (1,103g and 1,011g)

 - Line 291: Providing values for the 2 best accessions is pointless if he reader cannot compere it against the trail average or some kind of control value. It would be nice to know how much better these 2 accessions performed compared to the trial average. You must describe in your results section how your accessions performed for each trait: what was the range for each trait? Minimum, maximum and average accession values? Same for harvest index in line 297.

 - You need to indicate accessions that have promising traits to include in a breeding programme for the Andean-Amazonia region. Which accessions meet the criteria that a good variety for this region should have? Which accessions have traits that make them interesting to include in a breeding programme in crosses? 

Discussion
************
 - Add discussion on variability in the trial blocks and the weather conditions during the trial in comparison to long-term average. 

 - Discussion can be improved. Now, only correlations between productivity traits and yield are only briefly discussed for some traits. It would be better to also discuss relations between other productivity traits observed and yield :
 * earliness (D50F and D50M) vs. yield
 * W100 (preferably expressed as TSW) vs. yield
 * HI vs. yield
I also suggest to mention the actual correlation coefficients in the discussion text. 

 - References [58-59] are papers on rice, not quinoa. Rice is a monocot crop in which the relationship between plant architecture and yield may be very different than quinoa, so these references may not be very relevant for the present study. You need to refer to quinoa papers that found that panicle dimensions are associated with seed yield. Instead of the references on rice, it would be better to include more reference on recent studies in quinoa. In addition to the studies used [5] [25] [62] [63], it would be interesting to include this recent study published in Plants on quinoa in NW Europe: https://doi.org/10.3390/plants14010003 , and compare your correlation results to it. Some of your findings are similar (e.g. the strong relation between earliness and yield - now poorly discussed), some different (which may be explained by the different germplasm used). 

 - Limitations of the study: Additional important limitations that must be mentioned is that (1) you only performed this study in 1 growing season (with normal/dry/warm/??? [to describe] weather conditions compared to the long term average). Performance of some accessions may be different in different growing seasons due to different weather conditions. (2) There was very limited replication in the trial with augmented design: accessions were not repeated. In case of high variability between blocks, this leads to possibly distorted values for productivity traits in accessions with extreme performance. In that case, results should be interpreted very carefully. 

 - Line 432-433: which native accessions exactly will you recommend to use more in te Andean-Amazonia region? Or use in breeding intiatives? What benefits exactly will these accessions bring to the farmer? In which traits are they better than the quinoa varieties that the farmers are growing now? Or are they not yet growing quinoa in that region? 

Author Response

REVIEWER 1

I believe this manuscript is generally well-written and contains interesting information. English language is very good and the paper is written clearly and understandable. 

However, the manuscript must be improved before it can be accepted for publication. 

Response: Dear Reviewer, thank you very much for your comments and for the feedback aimed at improving the manuscript. Each of the suggestions will be addressed below, and all changes made to the document have been highlighted in red to facilitate your review.

Comments 1: This was a trial with augmented design in only 1 trial season; fully replicated design would have been better. Variation between blocks should be assessed and discussed. Another major remark is that the actual results obtained in this study for each accession are not presented. Without presenting this information, the significance of the manuscript is low, and the manuscript should in my opinion be rejected. Promising accessions should be mentioned. 

Response 1: 

We appreciate the comment and fully agree with the observation. In the Amazonas region, which corresponds to the Andean–Amazonian agroecological zone, quinoa is not an native crop. In this regard, the present study represents the first formal introduction of the crop in this region. We used an augmented design because the seeds used for the adaptation trial were donated by the Germplasm Bank of the Universidad Nacional de San Cristóbal de Huamanga (UNSCH); however, the quantity available was limited, which made it impossible to implement an experimental design with replications. Furthermore, the main objective of the study was to introduce and establish the crop under the Andean–Amazonian agroecological zone, conduct their agro-morphological characterization, and increase seed to ensure their conservation and future evaluation in replicated experiments. This information was incorporated into the Materials and Methods section (Lines 133–136), while the clarification regarding the absence of replications was included in the Study Limitations section (Lines 581– 601).

The complete data corresponding to the 158 accessions were incorporated into the M&M section (Line 108), complemented by a supplementary table detailing the passport information of each entry, including: UNTRM code, UNSCH code, agroecological zone, collection date, country, department, province, district, locality, and altitude (m.a.s.l.). In addition, the qualitative and quantitative phenotypic characterization of all accessions was included (Supplementary Table 1). To standardize material identification, the accessions were renamed following the quinoa crop ontology coding system, assigning the prefix 367 and a sequential number according to the order of entry (https://cropontology.org/page/CropCodes).

Below you can find my detailed comments. Please address each comment. Additional small textual edits can be found in the attached document. 

General

Comments 2: I have a problem with the numerous abbreviations that are used in this study. A lot of abbreviations are defined that are not always so straithforward. For the reader it is impossible to remember all abbreviations. However, abbreviations are often written in full again at multiple places in the text. E.g. productivity traits are repeated have been introduced in the M&M and are repeated in line 234-244, and again in line 265-275. Also they are repeated in figures and Tables. Do we really need all these abbreviations then? I suggest to limit the abbreviations and write them in full as much as you can. 

Response 2: We appreciate the observation. In response, we reduced the abbreviations to only those strictly necessary and restored in full any terms whose abbreviations did not contribute to clarity. In the M&M (Lines 161-186). At Results sections, we removed repeated definitions so that each abbreviation is introduced only once. Additionally, we standardized the accession codes following the quinoa crop ontology and updated the figures and tables to improve the overall readability of the manuscript.

Introduction

Comments 3: There are 5 types of quinoa, including the Altiplano type, valley type and the Yungas type. Given the environment if the study site, I would expect that yungas type or valley type performs best in this region. It would be good to add this information in the introduction. You can then use it again in the discussion to explain why some of the check cultivars (Altiplano types) did not perform well in this environment (line 428 - 430). 

Response 3: We appreciate the observation, and the recommendation has been incorporated. Information on the five types of quinoa was added to the revised Introduction (lines 57–68), and the list of accessions, along with their place of origin and agroecological zone, was also included (Supplementary Table 1).

Reference: 

Instituto Nacional de Innovación Agraria (INIA); Organización de las Naciones Unidas para la Alimentación y la Agricultura (FAO) Catálogo de variedades comerciales de quinua en el Perú; INIA & FAO: Lima, Perú, 2013.  

Maamri, K.; Zidane, O.D.; Chaabena, A.; Fiene, G.; Bazile, D. Adaptation of Some Quinoa Genotypes (Chenopodium Quinoa Willd.), Grown in a Saharan Climate in Algeria. Life 2022, 12, 1854, doi:10.3390/life12111854.

Soto, E.; Mercado, W.; Estrada Zúniga, R.; Díaz, F.; Díaz, G. El mercado y la producción de quinua en el Perú; Instituto Interamericano de Cooperación para la Agricultura – IICA Instituto Nacional de Innovación Agraria - INIA, 2015; ISBN 978-92-9248-602-0.

Comments 4: line 58-59: 2x erosion in the same sentence.  Try to find a different phrasing, e.g. loss of traditional cultivation practices.

Response 4: Thank you for pointing this out. We agree with the reviewer’s comment. Accordingly, the requested modification has been incorporated into the revised manuscript (lines 68–71).

Comments 5: line 89-91: please provide total annual precipitation (mm) for the trial region. 

Response 5: Thank you for pointing this out. We agree with the reviewer’s comment. The requested information has been incorporated into the revised manuscript (Lines 100–101).

Comments 6 : One of the objectives of this study is to identify accessions with potential for breeding in de Andean-Amazonian region (line 78-80). This implies that you need to describe what traits are important for that region. What kind of varieties are farmers growing in this region now, and what would we need for this region? Do the farmers in that region need early or late maturing varieties? tall plants or short? big os small seeds? Is susceptibility to downy mildew an issue? … Please describe which traits a suitable variety for that region would have. Then later in the results section you can indicate accessions that meet those criteria. 

Response 6: We appreciate the comment. This study represents the first introduction of quinoa into the Andean–Amazonian region, motivated by the need to diversify agricultural production and enhance food security in transitional Andean–Amazonian zones, where traditional crops may be affected by climate variability and environmental degradation. For these conditions, an appropriate ideotype was defined, combining high yield, short plant height to reduce lodging risk, earliness to align with the local agricultural calendar, resistance to pests and diseases, a single stem, and an amaranthiform panicle. The selection of amaranthiform panicles, as noted by Lozano-Isla et al. (2025), facilitates air circulation within the inflorescence, thereby decreasing the incidence of pests and diseases. This combination of traits reduces agronomic risks and provides a starting point for identifying promising accessions that may adapt and be used in breeding programs in the Andean–Amazonian region. This information has been incorporated into the Discussion section (Lines 528–537). The accessions that meet this criterion were included in the Results (Lines 364–366).

M&M

Comments 7: line 93-96 : Do you include in your manuscript a (Supplementary) Table with identification of the plant material (accession name, UNSCH reference number, accession properties) ? That would be an added value for the paper. 

Response 7: We appreciate the observation. Supplementary Table 1 has been incorporated with the complete identification details of the plant material, and its corresponding reference has been included in the revised manuscript (Line 108).

Comments 8: Line 100: a trial plan figure with organisation of the trial would be nice. Where were the control accessions placed in every block? Name your blocks from 1 to 16.

Response 8: We appreciate the observation. The experimental design was described in detail in  Supplementary Figure 1, which shows the overall layout of the trial, the placement of the control accessions within each block, and the numbering of the sixteen blocks. In this figure, codes “1,” “2,” and “3” represent the replicated check treatments: code 1 corresponds to INIA 415 Pasankalla, code 2 to INIA 420 Negra Collana, and code 3 to Blanca Juli. This information was included in Line 120-124. .

Comments 9: line 120-122: please provide total units of NPK fertilization/ If I calculate well, the trial received 118 units N, 78 units P and 0 units K units. N and P fertilization seems very adequate to me. Why no potassium fertilization? Does quinoa not use potassium?! 

Response 9: Thank you for the comment. Potassium is essential for quinoa; however, K was not applied due to its high availability in the soil (278.77 ppm), which is sufficient to meet the crop requirements (Supplementary Figure 2). This decision follows the nutrient management approach recommended by Lozano-Isla et al. (2020) and the technical sheet for the Salcedo INIA variety, which allows omitting Kâ‚‚O when soil levels are adequate (Lines 144–146).

Referencia:

Lozano-Isla, F.; Apaza, J.-D.; Mujica Sanchez, A.; Blas Sevillano, R.; Haussmann, B.I.G.; Schmid, K. Enhancing Quinoa Cultivation in the Andean Highlands of Peru: A Breeding Strategy for Improved Yield and Early Maturity Adaptation to Climate Change Using Traditional Cultivars. Euphytica 2023, 219, 26, doi:10.1007/s10681-023-03155-8.

Comments 10: line 161: For small-seed crops like quinoa, 1000-seed weight (TSW) is more commonly used than 100-seed weight. The used method to weight 1x 100 seeds is OK, but I strongly recommend to present the data as 1000-seed weights (just multiply by 10). The numbers will then also be larger than 1, making them easier to present. Use the abbreviation TSW instead of W100: TSW is commonly used. 

Response 10: Thank you for the comment. Changes have been made to the agromorphological descriptors related to seed weight, and they can now be viewed in the corresponding lines (Line 184-185) and in Supplementary Table 2.

Comments 11: line 162: I suggest to define easier abbreviations for SW10P and W10P. it is well described in the M&M that these traits were obtained using seeds from 10 plants, it is not necessary to include this information in the abbreviation. You may as well use 'SW' for seed weight and 'PAN' for panicle biomass. It will be easier to understand for the reader.

Response 11: Thank you for the suggestion. Changes have been made to the abbreviations, and they can be viewed in the corresponding lines (lines 185-191) and in the document. Included in the Supplementary Table 2.

Comments 12: line 189: this manuscript contains a lot of abbreviations. Abbreviations are useful only when they are used many times as they shorten the text. This abbreviation HCPC is only used once: just write it in full then please. Please also check for all other abbreviations in the text: if they are used only 3 times or less, just write them in full. 

Response 12: Thank you for pointing this out. We agree that abbreviations should be used only when necessary. The abbreviation HCPC was removed and written in full, and other infrequently used abbreviations were replaced with their complete terms in the revised manuscript (Lines 214).

Results

*********

Comments 13: The trial was organized in different blocks. The variability between the 16 blocks must be presented. You can again refer to the figure in the M&M.  In the discussion, you should then discuss the variability between blocks. Ideally, variation is low. In that case, BLUEs for the productivity traits for all accessions are close to the actually observed value. In case of high variability between the blocks, we can expect large adjustments to the BLUEs, which often causes distortion (shrinkage to the block mean) in productivity traits. In that case, the productivity traits obtained in your study for the accessions should be interpreted more carefully. This must be assessed and discussed. 

Response 13: We appreciate the observation. In the revised version, we incorporated the estimation of variability among the 16 blocks, and the detailed analysis is presented in ESM 1, where it can be seen that the variance attributable to the block effect was low. This reduced variability implies that the BLUEs for the productivity traits remained very close to the observed values, minimizing the risk of distortion due to shrinkage toward the block mean. This information was added to the Results section (Lines 307–313) and is not discussed further because, given the low variation among blocks, the inferences regarding accession performance can be considered robust.

Comments 14: It would be an added value if you could present the weather conditions (daily average temperature and daily precipitation for the growing seasons during the trial, and compare this to the long term average. This would allow you to discuss if the trial year was a dry, normal or wet year, and if it was warm, normal or cold. This is important information to explain why some accessions performed well or less well than expected. 

Response 14: The daily variation in temperature and precipitation was incorporated in the revised manuscript (lines 220–232).

Comments 15: A major remark is that the actual results obtained in this study for each accession are not presented. I expect to see 1 or 2 (Supplementary) Table(s) with the results for all the 12 quantitative and 11 qualitative traits for each accession. This is important information for readers that are searching for an accession with a specific trait to use in their own study. Without presenting this information, the manuscript has only little value to the broader audience and should in my opinion be rejected.  

Response 15:  Thank you for your comment. The complete data for the 158 accessions have been included in Supplementary Table 1 (Line 366), which provides the qualitative and quantitative phenotypic characterization of all accessions, covering the 12 quantitative and 11 qualitative traits for each accession.

Comments 16: A table is presented with the general findings for the 11 qualitative traits observed in this study. You also need to make a similar Table in the main document with the results for the quantitative traits: min and max for each trait, average, coefficient of variation (CV%), … This will help the reader to grasp the diversity present in the screened set, and compare the values obtained in your study to other quinoa screening studies. E.g. was the seed yield, plant height, ... obtained in your study higher or lower than in other studies? Now there is no way for the reader to assess this. 

Response 16:  We appreciate this observation and fully agree with the recommendation. In response, we incorporated a table summarizing the descriptive statistics (minimum, maximum, mean, and coefficient of variation) for all quantitative traits. This information has been included as Table 2 in the main document (Lines 285–305).

Comments 17: Line 234 - 244: abbreviations only need to be defined the first time they are introduced in the text. All abbreviations here already have been introduced in the M&M. No need to do it again here. Idem line 265-275. For figures and Tables (eg Figure 3) it is better to define the abbreviations again in the legend, as you do. 

Response 17: Thank you for the comment. Abbreviations are defined only the first time they are introduced in the text, as done in Materials and Methods. Therefore, they are not repeated elsewhere in the manuscript. As suggested, abbreviations have been redefined in the figure and table legends.

Comments 18: line 246-248: 1 digit after decimal point is enough for the % of variance explained: 35.1% and 16.0%. 

Response 18: Thank you for the comment. Your recommendation has been accepted, and the value has been adjusted in the manuscript (Lines 316-317).

Comments 19: line 291: (1,103.35 g and 1,010.82 g). Do we need the digits after the decimal point?? --> I suggest (1,103g and 1,011g)

Response 19: Thank you for the suggestion. Your recommendation has been accepted, and the values have been adjusted in the manuscript (line 391-395).

Comments 20: Line 291: Providing values for the 2 best accessions is pointless if he reader cannot compere it against the trail average or some kind of control value. It would be nice to know how much better these 2 accessions performed compared to the trial average. You must describe in your results section how your accessions performed for each trait: what was the range for each trait? Minimum, maximum and average accession values? Same for harvest index in line 297.

Response 20: We appreciate this comment. In the revised version, we addressed this concern by incorporating a table that reports the minimum, maximum, mean, and coefficient of variation for all quantitative traits. This information is now presented in Table 2 (Line 301), allowing a direct comparison between the two best-performing accessions, the trial average, and the full range of values observed across the panel. This addition provides a clearer context for interpreting how much better these accessions performed relative to the overall dataset. Individual accession-level data remain available in Supplementary Table 1. Additionally we made a comparison of the result with other studies (line 551-559).

Comments 21: You need to indicate accessions that have promising traits to include in a breeding programme for the Andean-Amazonia region. Which accessions meet the criteria that a good variety for this region should have? Which accessions have traits that make them interesting to include in a breeding programme in crosses? 

Response 21: Thank you for this comment. The identification of accessions with promising characteristics for breeding programs in the Andean–Amazonian region was carried out by integrating both quantitative and qualitative agromorphological traits, considering productive performance, tolerance to biotic factors, and efficiency in biomass-to-grain conversion. In the Discussion section, we propose a quinoa ideotype for this region based on these criteria (Lines 543–552). The accessions that meet the ideotype characteristics and show the greatest potential for inclusion in breeding programs or crossing schemes are highlighted in Supplementary Table 2 and in Group 2 of the principal component analysis (Figure 4b).

Discussion

Comments 22: Add discussion on variability in the trial blocks and the weather conditions during the trial in comparison to long-term average. 

Response 22: Thank you for this comment. Information regarding block-to-block variability and its relationship with environmental conditions has been incorporated into the Results section (Lines 307-3013). In addition, we expanded the Limitations section to clarify the rationale for using an augmented design as an appropriate tool for evaluating genetic diversity and large numbers of genotypes (Zystro et al., 2018; Burgueño et al., 2018). We also note that future studies will include replicated trials to enable a more detailed assessment of genotype-by-environment interactions (Lines 594-605).

References
Zystro, J.; Colley, M.; Dawson, J. (2018). Alternative Experimental Designs for Plant Breeding. In: Plant Breeding Reviews (Goldman, I., Ed.). Wiley. pp. 87–117. ISBN 978-1-119-52131-0.

Burgueño, J.; Crossa, J.; Rodríguez, F.; Yeater, K.M. (2018). Augmented Designs—Experimental Designs in Which All Treatments Are Not Replicated. In: ASA, CSSA, and SSSA Books (Glaz, B., Yeater, K.M., Eds.). American Society of Agronomy, Crop Science Society of America, and Soil Science Society of America, Madison, WI, USA. pp. 345–369. ISBN 978-0-89118-360-0.

Comments 23: Discussion can be improved. Now, only correlations between productivity traits and yield are only briefly discussed for some traits. It would be better to also discuss relations between other productivity traits observed and yield :

 * earliness (D50F and D50M) vs. yield

 * W100 (preferably expressed as TSW) vs. yield

 * HI vs. yield

I also suggest to mention the actual correlation coefficients in the discussion text. 

Response 23: Many thanks for this valuable suggestion. We revised and expanded the Discussion section to include a more comprehensive analysis of the relationships between productivity traits and yield, incorporating earliness (DF and DM), thousand seed weight (TSW), and harvest index (HI). These traits were evaluated in the context of selecting promising accessions and in relation to the proposed quinoa ideotype for the Andean–Amazonian region. In addition, the corresponding correlation coefficients were included to support the interpretation of these relationships. This information was added in Lines 500–540.

Comments 24: References [58-59] are papers on rice, not quinoa. Rice is a monocot crop in which the relationship between plant architecture and yield may be very different than quinoa, so these references may not be very relevant for the present study. You need to refer to quinoa papers that found that panicle dimensions are associated with seed yield. Instead of the references on rice, it would be better to include more reference on recent studies in quinoa. In addition to the studies used [5] [25] [62] [63], it would be interesting to include this recent study published in Plants on quinoa in NW Europe: https://doi.org/10.3390/plants14010003 , and compare your correlation results to it. Some of your findings are similar (e.g. the strong relation between earliness and yield - now poorly discussed), some different (which may be explained by the different germplasm used). 

Response 24: Thank you very much for the observation. The references have been updated, and the suggested article was included for additional support (Line 526-530).

Comments 25: Limitations of the study: Additional important limitations that must be mentioned is that (1) you only performed this study in 1 growing season (with normal/dry/warm/??? [to describe] weather conditions compared to the long term average). Performance of some accessions may be different in different growing seasons due to different weather conditions. (2) There was very limited replication in the trial with augmented design: accessions were not repeated. In case of high variability between blocks, this leads to possibly distorted values for productivity traits in accessions with extreme performance. In that case, results should be interpreted very carefully. 

Response 25: Thank you very much for the comments. We agree with the inclusion of these limitations. Additionally, information on the environmental conditions during the trial was expanded in the Results section (Lines 220–229), and the limitations of the experimental design, its application, and the rationale for its use were explained (Line 594-605).

Comments 26: Line 432-433: which native accessions exactly will you recommend to use more in te Andean-Amazonia region? Or use in breeding intiatives? What benefits exactly will these accessions bring to the farmer? In which traits are they better than the quinoa varieties that the farmers are growing now? Or are they not yet growing quinoa in that region?

Response 26: Thank you for the comment. Information on the traits relevant for selecting quinoa in the Andean–Amazonian region was included in the Discussion section (Lines 543-552). Additionally, the selected lines are indicated (Lines 390–5419), and all detailed information on the accessions has been included in Supplementary Table 1.

Reviewer 2 Report

Comments and Suggestions for Authors
  1. Thank you for the nice work.
  2. Germplasm evaluation requires a multi-location or multi-year study for confirmation. A single-year and one-location study is not enough for any conclusion. It is better to re-evaluate at least a subset of the accession for confirmation. 
  3. A list of accessions, including ID/name, origin, ecotype, and genotype, is necessary. 

Author Response

REVIEWER 2

Comments 1: Thank you for the nice work.

Response 1:  We appreciate the comment and thank you for your positive evaluation of our work.

Comments 2:Germplasm evaluation requires a multi-location or multi-year study for confirmation. A single-year and one-location study is not enough for any conclusion. It is better to re-evaluate at least a subset of the accession for confirmation. 

Response 2: We appreciate the observation and agree that validating the performance of germplasm requires multilocation trials to reach definitive conclusions. However, it is important to note that quinoa is not native to the Andean–Amazonian region, and therefore this study represents the first introduction and evaluation of the crop in this agroecological zone. In this context, the results should be understood as an initial exploratory assessment, providing baseline information on the performance of these 158 accessions.

Additionally, since this was the first planting in the area and the seeds were received as donations in very limited quantities, an augmented design was necessary to multiply seed and generate sufficient material to establish, in later stages, multi-environment trials (METs) that will allow a more robust validation of germplasm performance. This information was included in the Materials and Methods section (Lines 133–136).

This information is valuable for guiding future evaluations across multiple environments, as well as for identifying accessions with adaptation potential that can later be incorporated into breeding programs focused on the region. This clarification has been included in the Limitations and Future Perspectives section of the study (Lines 595-605) and includes the perspectives from Line 606-615.

Comments 3:A list of accessions, including ID/name, origin, ecotype, and genotype, is necessary. 

Response 3: Thank you for the suggestion. Complete information on the accessions, including ID/name, origin, ecotype, and genotype, has been included in Supplementary Table 1, providing a detailed record for each entry.

Reviewer 3 Report

Comments and Suggestions for Authors
  1. Lines 200-229: The results for He, H', and fi are described with three decimal places in the text, while the corresponding results in the table are shown with two decimal places. Should these be standardized?
  2. Lines 200-205: The statistical results for the "panicle color at 50% flowering" trait in Table 1 are unrelated to Figure 2a.
  3. Figure 2a: The image and its caption do not clearly label the traits corresponding to each variety, such as color, panicle type, etc. (The discussion mentions the traits and density of the panicles in Figure 2a. It is suggested that the image should label the panicle type and density for each variety. The discussion also mentions the seed color and shape in Figure 2b, but Figure 2b does not show the seed shape clearly.)
  4. Figure 2b: (1) Seed coat color trait—It is suggested to label the seed color in the figure. The seed color is not uniform. For example, UNTRM121014 appears to have purple, milky white, and gray colors. What is the standard for seed coat color when performing statistical analysis?
  5. Table 1: The table format is incorrect.
  6. 161 quinoa varieties were analyzed for traits, but 24 quinoa varieties were not included due to growth issues. Therefore, Figure 4 should reflect statistical analysis for 137 quinoa varieties. Line 265 in the text mentions 136 quinoa varieties.
  7. Line 266: PH and D50F show r = 0.79, but the corresponding data in the figure is 0.55. Similarly, D50PM and W10P show r = 0.53, while the figure data is 0.57, etc. Please carefully verify all other related data.
  8. Lines 290-292: The data analysis is unrelated to Figure 3b.
  9. High-yield standard: The final conclusion identifies four quinoa varieties as early-maturing and high-yielding. Is there a standard definition for high yield, or does it simply refer to varieties with higher yields among the 137 quinoa varieties? Is there a unified standard for defining high-yield?

Author Response

REVIEWER 3

Comments 1: Lines 200-229: The results for He, H', and fi are described with three decimal places in the text, while the corresponding results in the table are shown with two decimal places. Should these be standardized?

Response 1: Thank you for the comment. All values for He, H′, and fi have been standardized and are now presented consistently with two decimal places in both the text and the tables (Line 242-275). Also the Table 1 was updated to show only two decimals.

Comments 2: Lines 200-205: The statistical results for the "panicle color at 50% flowering" trait in Table 1 are unrelated to Figure 2a.

Response 2: Thank you for the comment. The reference in the text regarding the trait “panicle color at 50% flowering” has been removed to maintain consistency with the corresponding figure (Line 247).

Comments 3: Figure 2a: The image and its caption do not clearly label the traits corresponding to each variety, such as color, panicle type, etc. (The discussion mentions the traits and density of the panicles in Figure 2a. It is suggested that the image should label the panicle type and density for each variety. The discussion also mentions the seed color and shape in Figure 2b, but Figure 2b does not show the seed shape clearly.)

Response 3: Thank you for the comment. To improve visualization and maintain clarity, Figure 3 was updated and now presents the phenotypic diversity in panicle traits and seed episperm color: While only seed color is shown in Figure 3c, other traits mentioned in the discussion, such as seed shape and panicle density, are presented and analyzed in detail in the tables and text (Lines 242-275).

Comments 4: Figure 2b: (1) Seed coat color trait—It is suggested to label the seed color in the figure. The seed color is not uniform. For example, UNTRM121014 appears to have purple, milky white, and gray colors. What is the standard for seed coat color when performing statistical analysis?

Response 4: Thank you very much for the observation. To improve the understanding of variability and seed episperm color in quinoa, the images in Figure 3c were updated. To categorize the seed episperm color we used the descriptors for quinoa (Bioversity International et al., 2013), and this information is also provided in Supplementary Table 2 (Line 158).

Reference:

Bioversity International; Organización de las Naciones Unidas para la Agricultura y la Alimentación (FAO); Fundación PROINPA; Instituto Nacional de Innovación Agropecuaria y Forestal (INIAF); Fondo Internacional de Desarrollo Agrícola (FIDA) Descriptores para quinua (Chenopodium quinoa Willd.) y sus parientes silvestres; Bioversity International, 2013; ISBN 978-92-9043-927-1.

Comments 5: Table 1: The table format is incorrect.

Response 5: Thank you for the comment. The format of Table 1 has been corrected and updated according to the manuscript guidelines (Line 275).

Comments 6: 161 quinoa varieties were analyzed for traits, but 24 quinoa varieties were not included due to growth issues. Therefore, Figure 4 should reflect statistical analysis for 137 quinoa varieties. Line 265 in the text mentions 136 quinoa varieties.

Response 6:  We appreciate this observation and apologize for the error. Out of the initial 161 accessions, 158 were evaluated, as three were used as controls to adjust the design and obtain the BLUEs according to the augmented design (Lines 120-124). Of these, 24 accessions did not complete their development. Both Figure 4 and the text have been corrected to accurately reflect the number of accessions evaluated, which was 134 (Line 240). Detailed information on each accession and their phenotypic evaluations is provided in Supplementary Table 1.

Comments 7: Line 266: PH and D50F show r = 0.79, but the corresponding data in the figure is 0.55. Similarly, D50PM and W10P show r = 0.53, while the figure data is 0.57, etc. Please carefully verify all other related data.

Response 7: Thank you very much for the observation. All correlation data were carefully reviewed and corrected in both the text and figures to ensure consistency (Lines 350–369).

Comments 8: Lines 290-292: The data analysis is unrelated to Figure 3b.

Response 8: Thank you for the observation. The information has been corrected, and the cross-reference has been properly adjusted (Line 381).

Comments 9: High-yield standard: The final conclusion identifies four quinoa varieties as early-maturing and high-yielding. Is there a standard definition for high yield, or does it simply refer to varieties with higher yields among the 137 quinoa varieties? Is there a unified standard for defining high-yield?

Response 9: We appreciate the comment. In this study, no universal high-yield standard was used; the classification was primarily based on a relative comparison among the 134 accessions evaluated under the specific conditions of the Andean–Amazonian region. Accessions identified as high-yielding were those that exceeded the average performance of the evaluated set (Table 2 - Line 301) and also met the criteria established for the proposed ideotype for this region: earliness, short plant stature, resistance to pests and diseases, single stem, and amaranthiform panicle (Lines 543-552).

To strengthen the interpretation, a contextual comparison was also performed with values reported in other studies on quinoa yield and adaptation in different production regions (Lozano-Isla et al., 2023; Vleugels et al., 2025). This comparison confirmed that the selected accessions show competitive yield levels even when external references are considered. The discussion of these aspects has been incorporated into the Discussion section of the manuscript (Lines 553-558).

References:

Lozano-Isla, F.; Apaza, J.-D.; Mujica Sanchez, A.; Blas Sevillano, R.; Haussmann, B.I.G.; Schmid, K. Enhancing Quinoa Cultivation in the Andean Highlands of Peru: A Breeding Strategy for Improved Yield and Early Maturity Adaptation to Climate Change Using Traditional Cultivars. Euphytica 2023, 219, 26, doi:10.1007/s10681-023-03155-8.

Vleugels, T.; Van Waes, C.; De Keyser, E.; Cnops, G. Optimization of Breeding Tools in Quinoa (Chenopodium Quinoa) and Identification of Suitable Breeding Material for NW Europe. Plants 2025, 14, 3, doi:10.3390/plants14010003.

Round 2

Reviewer 1 Report

Comments and Suggestions for Authors

Dear authors, 
Dear editor, 

I believe the manuscript has greatly improved after the revisions. However, a number of minor edits remain to be implemented. 
You can find my remarks below, and additional textual suggestions in the attached document. 

When these issues have been resolved, I believe the manuscript can be accepted for publication. 

Supplementary Tables and figures
**********************************
ESM_3: some spanish words need to be translated in english:
columns PCF (purpura, mixtura, …), PDE (laxa, intermedia), GS (cilíndrico, lenticular). Please check the entire table and put everything in english.

Sup Figure 2 with the soil analysis report can be omitted: it is not necessary. I am satisfied with the added text in line 144-146. There was enough K in the soil so you did not apply any extra K. Please write 274 ppm and not 274.47 ppm, see my remarks in previous revisions!

Main manuscript
****************
line 115-120 + Sup Figure 1: The organisation of blocks in the trial is still unclear. In the text you write that there were 161 quinoa accessions tested and that the trail was divided into 16 blocks (line 116). In the text you also describe that each block was divided into 13 columns and 16 rows (line 118). Then the question is: does Supp Figure 1 indicate the lay-out of the total trial, or of 1 block? Sup Figure 1 does show 161 accessions, so I guess that it shows the full trial? If so, you need to indicate the 16 'blocks' in the figure. How did you divide the trial into blocks? Needs clarification in the Figure and text!

Line 241 + Sup Table 1: The text and table describes 158 accessions. There were 161 accession in the trial: 3 checks + 158 test accessions. Why don't you provide the data for the check accessions in Sup Table 1? Then the reader can compare the performance of your test accessions against the check accessions. 

line 282-284: 24 accessions did not perform well and were excluded from the 'final analysis'. Does this mean you did not include these accessions in the PCA and clustering analysis? Then please also indicate it in the M&M section where you describe the PCA and clustering. 

line 285 - 300 + Table 2: please reduce numbers after decimal point to 1 (for numbers > 10) or 2 (for numbers between 0 and 10) everywhere. Wee my remark in previous revision.

Line 307: it is still unclear how the 16 blocks were designed in the study. This must be improved. See previous remark. 

line 388: 'genotypes' and 'accessions' cannot be used interchangeably in quinoa. Quinoa is not a 100% self-pollinating species. Therefore, quinoa accessions are not 100% homogeneous and can contain different genotypes. In this study you tested accessions: you should call the accessions  (or varieties or populations) but not genotypes. Please check the entire manuscript and change. 

line 420- 426: this paragraph does not belong here, it would fit better directly after line 282-284: These accessions were not included in the clustering so it is better to describe them before the PCA and clustering results. Please integrate it in the text there. 

line 429-430: data for the three control cultivars are not given: not in the Sup Table 1, not in the next nor anywhere else. Why? 

line 478-479: do you have information on seed saponin content for the varieties tested in your study? If so it would be nice to know the correlation with Epicauta incidence. If not, you can leave the text as it is. 

Author Response

REVIEWER 1

I believe the manuscript has greatly improved after the revisions. However, a number of minor edits remain to be implemented.
You can find my remarks below, and additional textual suggestions in the attached document. 

When these issues have been resolved, I believe the manuscript can be accepted for publication. 

Response: I appreciate the comments provided and the thorough review of the manuscript. All the observations noted, including the textual suggestions in the attached document, were reviewed and corrected in the current version.

Supplementary Tables and figures
**********************************
Comments 1: ESM_3: some spanish words need to be translated in english:
columns PCF (purpura, mixtura, …), PDE (laxa, intermedia), GS (cilíndrico, lenticular). Please check the entire table and put everything in english.

Response 1: We appreciate the observation. The corrections indicated for Table ESM_3 have already been made in the current version of the supplementary material.

Comments 2: Sup Figure 2 with the soil analysis report can be omitted: it is not necessary. I am satisfied with the added text in line 144-146. There was enough K in the soil so you did not apply any extra K. Please write 274 ppm and not 274.47 ppm, see my remarks in previous revisions!

Response 2: Thanks for the suggestion. Supplementary Figure 2 has been removed and the text has been adjusted, including the correction of the potassium value to 274 ppm (Lines 147–148).

Main manuscript
****************
Comments 3: line 115-120 + Sup Figure 1: The organisation of blocks in the trial is still unclear. In the text you write that there were 161 quinoa accessions tested and that the trail was divided into 16 blocks (line 116). In the text you also describe that each block was divided into 13 columns and 16 rows (line 118). Then the question is: does Supp Figure 1 indicate the lay-out of the total trial, or of 1 block? Sup Figure 1 does show 161 accessions, so I guess that it shows the full trial? If so, you need to indicate the 16 'blocks' in the figure. How did you divide the trial into blocks? Needs clarification in the Figure and text!

Response 3: Thank you for the comment. In this type of design, FieldHub defined the blocks strictly through statistical procedures rather than as geometric subdivisions of the field. Consequently, the 16 blocks generated in the study corresponded directly to the 16 horizontal rows of the 16 × 13 experimental matrix, with each row acting as an independent block that included one replication of the three check cultivars together with the unreplicated accessions. This configuration allowed the checks to function as internal controls to adjust for the environmental heterogeneity present in the field, which constitutes the fundamental principle of augmented designs (Lines 117–122).

Comments 4: Line 241 + Sup Table 1: The text and table describes 158 accessions. There were 161 accession in the trial: 3 checks + 158 test accessions. Why don't you provide the data for the check accessions in Sup Table 1? Then the reader can compare the performance of your test accessions against the check accessions. 

Response 4: We appreciate the observation. The data corresponding to the three check materials have already been incorporated into Supplementary Table 1, so the trial information is now complete and allows for the requested comparison.

line 282-284: 24 accessions did not perform well and were excluded from the 'final analysis'. Does this mean you did not include these accessions in the PCA and clustering analysis? Then please also indicate it in the M&M section where you describe the PCA and clustering. 

Response: The modification has already been incorporated. It was clarified that the 24 accessions were excluded from the final analysis and were not included in the PCA or the clustering analysis. This clarification has been indicated in the Materials and Methods section (Lines 115 -117).

line 285 - 300 + Table 2: please reduce numbers after decimal point to 1 (for numbers > 10) or 2 (for numbers between 0 and 10) everywhere. Wee my remark in previous revision.

Response: We appreciate the observation. The table was modified to separate the units and standardize the number of decimal places, and the corresponding text was also adjusted as indicated (Lines 295–312).

Line 307: it is still unclear how the 16 blocks were designed in the study. This must be improved. See previous remark. 

Response: Thank you for the observation. The organization of the 16 blocks has already been clarified in the manuscript and responded above (lines 117–122).

line 388: 'genotypes' and 'accessions' cannot be used interchangeably in quinoa. Quinoa is not a 100% self-pollinating species. Therefore, quinoa accessions are not 100% homogeneous and can contain different genotypes. In this study you tested accessions: you should call the accessions  (or varieties or populations) but not genotypes. Please check the entire manuscript and change. 

Response: We appreciate the observation. Indeed, because quinoa is not a fully self-pollinating species and accessions may contain multiple genotypes, the term “genotype” is not appropriate to describe the material evaluated. Therefore, the entire manuscript was revised, and all instances of “genotypes” were replaced with “accession” as appropriate.

line 420- 426: this paragraph does not belong here, it would fit better directly after line 282-284: These accessions were not included in the clustering so it is better to describe them before the PCA and clustering results. Please integrate it in the text there. 

Response: Thank you very much for the observation. The relocation has already been carried out, and the paragraph is now on lines 287–293.

line 429-430: data for the three control cultivars are not given: not in the Sup Table 1, not in the next nor anywhere else. Why? 

Response: We appreciate the observation. The data for the three control varieties have already been added to Supplementary Table 1.

line 478-479: do you have information on seed saponin content for the varieties tested in your study? If so it would be nice to know the correlation with Epicauta incidence. If not, you can leave the text as it is. 

Response: Thanks for the observation. The saponin content analysis was not conducted on the evaluated varieties, so it was not possible to estimate its correlation with the incidence of Epicauta sp. This analysis will be addressed in future studies; therefore, the text was left as originally written. Thanks.

Reviewer 2 Report

Comments and Suggestions for Authors

Thank you for clarification. Please accept. Thank you. 

Author Response

Comment: Thank you for clarification. Please accept. Thank you. 

Response: We sincerely appreciate your valuable comments and suggestions, which have contributed to improving the quality and clarity of the manuscript. Thank you for your time and consideration.

Reviewer 3 Report

Comments and Suggestions for Authors

The author has fully addressed my questions and the manuscript is recommended for publication.

Author Response

Comment: The author has fully addressed my questions and the manuscript is recommended for publication.

Response: We sincerely appreciate your valuable comments and suggestions, which have contributed to improving the quality and clarity of the manuscript. Thank you for your time and consideration.